

# The status quo and influencing factors of breastfeeding social support in China based on LASSO regression model

Tian Sun, Chanzhi Duan, Yan Wang and Qing Li

School of Nursing, Chengde Medical University, Chengde, Hebei Province, China

## ABSTRACT

**Background**. In China, the rate of exclusive breastfeeding at 6 months is only 29.2%, well below the global breastfeeding collective target of at least 50% by 2025. This study explores the status quo of breastfeeding social support among *puerpera* and analyses its influencing factors, in order to provide a basis for improving breastfeeding rate in China.

**Methods**. A total of 251 *puerpera* were selected to fill in the general information questionnaire and the Breastfeeding Social Support Scale from three community health service centers in the city of Chengde, Hebei province, China, from September to November 2023 by completed random cluster sampling method. And the LASSO regression was used to screen and order the influencing factors of breastfeeding social support.

**Results**. The total score of the Breastfeeding Social Support Scale was (48.95 ± 7.93). The results of LASSO regression showed that when the lambda (λ) value was 0.7428, the error was the smallest, and the corresponding number of influencing factors was four, and the top four independent variables in importance ranking were whether or not breastfeeding at night, employment status, the feeding mode of the youngest baby, partner attended school/lecture for pregnant women.

**Conclusion**. The level of breastfeeding social support remains to be improved; health care providers were suggested to develop targeted intervention according to the influencing factors.

## BACKGROUND

Breastfeeding is not only beneficial for the health of both mothers and infants, but also contributing to sustainable development of society (*World Health Organziation, 2021a*). The World Health Organization (WHO), The Health Commission of the People's Republic of China (NHC), and other Health Alliance in the world recommend exclusive breastfeeding of infants under 6 months of age, followed by continued breastfeeding until at least 2 years old (*World Health Organziation, 2021b*; *National Health, Maternal and Child Development, 2021*). However, exclusive breastfeeding rates for the first 6 months remain not very optimistic in many regions worldwide.

Corresponding author
Qing Li, liqing8168@cdmc.edu.cn, liqing8168@126.com

This is the case in China, for example, although 60.8% of mothers during late pregnancy have the intention to exclusively breastfeed, and 58.3% know that their babies need to be exclusively breastfed within 6 months of birth, in fact only 29.2% can achieve, well below the global breastfeeding collective target of at least 50% by 2025 (*World Health Organziation, 2021a*; *Zhao et al., 2022*; *China Development Research Foundation, 2019*). The gap between early intention and later practice due to various causes, including maternal and infant factors, as well as social support from professionals, peers, and family members (*Prentice, 2022*). Among them, the availability of social support is the key to help bridge the gap. A study in Ethiopia found that emotional support and practical help for breastfeeding from partner were effective in increasing breastfeeding knowledge, and improving the breastfeeding attitude (*Shitu et al., 2021*). An intervention study in Chile showed that virtual support from peers and professional support from breastfeeding experts was raised the rate of exclusive breastfeeding in 6 months postpartum (*Lucchini-Raies et al., 2023*). Also, studies in Minnesota in the USA and in Australia revealed that providing women with telephone counseling, face-to-face meetings, and text messaging from trained peers may benefit the initiation and continuation of breastfeeding (*McCoy et al., 2018*; *Forster et al., 2019*). Therefore, it is essential to accurately measure the current level of breastfeeding social support and explore its influencing factors, so that we can develop the effective interventions.

Previous studies often applied univariate analysis and multivariate analysis to assess the correlated factors, but lack of the ranking among variables. The least absolute shrinkage and selection operator method (LASSO) is a new field in medical research that has gained popularity in recent years. It is a promising machine learning method to screen and order important variables from large amounts of data, for instance, a study in China used LASSO regression to identify the psychological capital influencing factors of patients with breast cancer (*Na et al., 2023*). In this study, we evaluated the level of breastfeeding social support and used LASSO regression to analyze its related factors among the *puerpera* in 2 years postpartum.

## MATERIALS & METHODS

### Study design and participants

This was a cross-sectional study that used completed random cluster sampling, the study was conducted in the city of Chengde, Hebei province, China. The sample size was estimated using the formula by *Riley et al. (2020)*. Considering a 95% confidence level ($\alpha = 0.05$), and a margin of error of ($d$) 0.05, the proportion of breastfeeding social support was based on a previous study conducted in China, which assessed the prevalence approximately 80% (*Yuan et al., 2024*), adding 5% for an incomplete rate, we finally determined the sample size was 258.

There are 15 community health service centers in Chengde, we referred to the sample size required for the study, results of the pre-survey and bulletin of the Seventh National Population Census of China, to determine how many health service centers we select. Mothers who visited the three community health service centers from September to

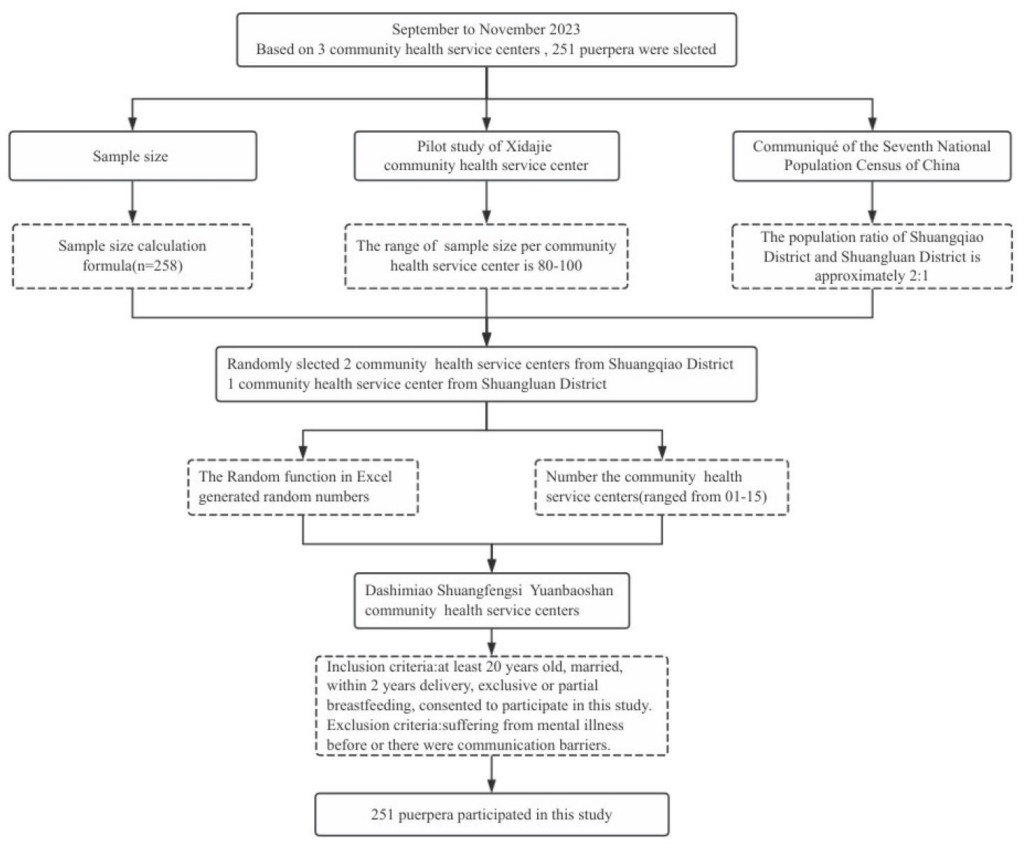

**Figure 1 Flowchart.**

November, 2023 for their infant's vaccine were invited to participate our study (Fig. 1), After the exclusion criteria described above were applied, 251 mothers participated in this study, and 241 (96.02%) completed the survey.

## General information questionnaire for *puerpera*

The questionnaire was designed by ourselves based on the comprehensive literature review and experts' advices, and it included questions such as age, nation, education level, occupation, average income per person in family and the most recent delivery mode.

## Breastfeeding social support scale

To measure the level of breastfeeding social support in *puerpera*, we used the Breastfeeding Social Support Scale (BSSS) (*Nanishi, Green & Hongo, 2021*). The scale includes 11 items with three dimensions: breastfeeding peers and named professions, practical help, and support to meet emotional needs. The items are answered using a Likert scale ranging from 1 to 5 (1 = inconsistent to 5 = consistent); the scores range between 11 and 55; the higher scores indicate more breastfeeding social support. The measure showed adequate reliability and validity. The Cronbach's $\alpha$ coefficient was 0.83 when measured 3 months postpartum and 0.85 when measured 5 months postpartum. The item-content validity ranged from
0.80~1.00, the scale-content validity was 0.90, and the exploratory factor analysis showed the 3-factor model was stable.

After authorization was obtained, we translated the BSSS into Chinese based on the principle of scale cross-cultural adaptation of the Guidelines recommended by the American Association of Orthopaedic Surgeons Committee on Evidence-Based Medicine, and invited six experts on the field of breastfeeding to assess the correlation between each item and its dimension, as well as its relevance to the breastfeeding social support. The item-content validity ranged from 0.830~1.000; the scale-content validity was 0.820. Before the formal investigation, we chose 30 *puerpera* who confirmed the inclusion criteria to conduct a preliminary test to check the reliability of the scale. The resulting data was statistically analyzed with the software SPSS 25.0 and AMOS 23.0. The Cronbach's $\alpha$ coefficient of the 11-item scale was 0.866; the test-retest reliability was 0.815, and the split-half reliability was 0.813. The structure of the Chinese version of BSSS differs slightly from the original scale, the three dimensions that have been modified are as follows. Support from health care providers (three items), practical help (three items), support from people the mother can rely on to help meet emotional needs (five items) (*Tian et al., 2024*). The Chinese version of BSSS was used in this study.

## Statistical analyses

The EpiData 3.1 software was used for data input, the R Studio. Ink (Vienna statistical computing foundation, Austria) and the glmnet package were used to carry out the lasso regression analysis. Statistical description was performed using SPSS 25.0 software (IBM, Armonk, NY, USA), with qualitative data expressed as frequencies and proportions, and quantitative data expressed as mean $\pm$ standard deviation if they followed a normal distribution. If not, median and interquartile range were used. Considering the number of items in each dimension is different, the range of score will also different. We used the average score of each dimension (the total score of the dimension/the number of items of the dimension) to order them.

## Ethical considerations

Ethical approval was obtained from the Medical Ethics Committee of Affiliated Hospital of Chengde Medical University (number: CYFYLL2022331), and written informed consent was obtained from all participants before collecting data.

## RESULTS

### Characteristics of participants

Characteristics of participants are shown in Table 1. Their average age was (31.01 $\pm$ 4.01) (range 20~45) years. Among the 241 participants, 92 (38.20%) aged between 20 and 29 years; 114 (47.30%) aged between 30 and 35 years; 35 (14.50%) 35 years or older. 155 mothers (64.30%) belonged to Han ethnicity, and an additional 86 (35.70%) belonged to other ethnicities. Moreover, most mothers breastfed at night (97.90%), while the rest were not (2.10%).

**Table 1 Characteristics of participants.**

| Variables | Cases | Proportion (%) |
|---|---|---|
| Age | | |
|     20∼29 | 92 | 38.20 |
|     30∼35 | 114 | 47.30 |
|     >35 | 35 | 14.50 |
| Ethnic groups | | |
|     The Han nationality | 155 | 64.30 |
|     The other nationality | 86 | 35.70 |
| Highest level of schooling | | |
|     Junior high | 33 | 13.70 |
|     High school | 37 | 15.40 |
|     College | 58 | 24.10 |
|     University | 93 | 38.60 |
|     Postgraduate or higher | 20 | 8.30 |
| Employment status | | |
|     Be unemployed | 75 | 31.12 |
|     Be employed | 166 | 68.88 |
| Financial status | | |
|     <3000 | 19 | 7.90 |
|     3000∼4999 | 74 | 30.70 |
|     5000∼7999 | 105 | 43.60 |
|     ⩾8000 | 43 | 17.80 |
| The most recent delivery mode | | |
|     Natural labor | 90 | 37.34 |
|     Caesarean section | 151 | 62.66 |
| Number of babies | | |
|     1 | 138 | 57.30 |
|     2 | 83 | 34.40 |
|     ⩾3 | 20 | 8.30 |
| Gender of the youngest baby | | |
|     Male | 112 | 46.50 |
|     Female | 129 | 53.50 |
| The age of the youngest baby | | |
|     0∼5 months | 144 | 59.80 |
|     6∼12 months | 72 | 29.90 |
|     13∼24 months | 25 | 10.40 |
| The feeding mode of the youngest baby | | |
|     Exclusive breastfeeding | 87 | 36.10 |
|     Mixed feeding | 154 | 63.90 |

**Table 1** (*continued*)

| Variables | Cases | Proportion (%) |
|---|---|---|
| The way of confinement | | |
| Family members | 149 | 61.60 |
| Confinement nurse | 65 | 26.90 |
| Confinement center | 26 | 10.70 |
| Nanny | 2 | 0.80 |
| Whether or not breastfeeding at night | | |
| Yes | 237 | 97.90 |
| No | 5 | 2.10 |
| Main caregiver of the youngest baby during the day | | |
| Grandmother/grandfather | 89 | 36.90 |
| Own | 132 | 54.80 |
| Father | 3 | 1.20 |
| Nanny/confinement nurse | 17 | 7.10 |
| Maternity leave time | | |
| No | 43 | 25.75 |
| Below statutory maternity leave | 28 | 16.77 |
| Statutory maternity leave(158~173 days) | 78 | 46.70 |
| Above statutory maternity leave | 18 | 10.78 |
| Return to work | | |
| Yes | 103 | 61.68 |
| No | 64 | 38.32 |
| Lactation room in the workplace | | |
| Yes | 25 | 24.27 |
| No | 78 | 75.73 |
| Mother attended school/lecture for pregnant women | | |
| Never | 190 | 78.50 |
| 1~3 | 40 | 16.50 |
| 4~6 | 10 | 4.10 |
| ≥7 | 2 | 0.80 |
| Partner attended school/lecture for pregnant women | | |
| Never | 213 | 88.00 |
| 1~3 | 26 | 10.70 |
| 4~6 | 2 | 0.80 |
| ≥7 | 1 | 0.40 |
| Community promotion of breastfeeding methods | | |
| Bulletin board | 87 | 36.00 |
| Phoned or texted or answered questions | 18 | 7.40 |
| Provided house calls | 5 | 2.10 |
| No | 132 | 54.50 |

**Notes.**
There are 3 skip questions in this questionnaire. The number of participants who filled in the maternity leave time, return to work and lactation room in the workplace were 168, 168, 104, respectively.

### The status quo of breastfeeding social support among the *puerpera* in 2 years postpartum

The total score of the Breastfeeding Social Support Scale was (48.95 ± 7.93). The scores of each dimension were as follows: practical help (13.52 ± 3.00), support from people the mother can rely on to help meet emotional needs (21.80 ± 4.20), support from health care providers (13.62 ± 2.45). And the average scores of the support from health care providers dimension (4.59 ± 0.81) were the highest, the middle was practical help dimension (4.51 ± 1.00), support from people the mother can rely on to help meet emotional needs (4.37 ± 0.84) were the lowest.

### Variable selection

All variables were entered into the LASSO regression with the log (lambda) value of the harmonic parameter, the importance of variables changed along with the change of the lambda, the assignments of the variables are showed in Table 2. When the lambda (λ) value was 0.7428, the error was the smallest, and the corresponding number of variables screened out by the LASSO regression is shown in Fig. 2. We built an influencing factor classifier by using the lasso regression (Fig. 3). After LASSO analysis, four influencing factors were selected, including whether or not breastfeeding at night, employment status, the feeding mode of the youngest baby, partner attended school /lecture for pregnant women (Table 3).

## DISCUSSION

In this study, the breastfeeding social support scale exhibited an overall score of (48.95 ± 7.93), which suggests the breastfeeding social support in Chengde city is positioned at an upper-middle level. The level is nearly as the same as a previous study that used Social Support Rating Scale to measure it (*Yanli, Nafei & Lan, 2022*). Furthermore, it surpassed the level in a study conducted by the Huges Breastfeeding Support Scale (*Yumei, 2020*). This suggests our breastfeeding promotion strategies has made some effects in recent years, but we still has great potentials for making further progress. We also found that the average scores of support from health care providers were the highest, and the average scores of support from people the mother can rely on to help meet emotional needs were the lowest. Which was consistent with the previous studies results in other countries, those studies indicated that healthcare professions have provided sufficient professional support and emotional support to breastfeeding mothers (*Rouse & Ferrarello, 2019*; *Alharthi et al., 2019*). However, a study conducted by *Temesgen et al. (2023)* showed that highest support was emotional support, and 45.8% *puerpera* cannot get practical help, among them 47.9% cannot get help when they were cooking, 61.8% cannot get help when they were doing the laundry. It is possible that we have improved the rights and interests protection for breastfeeding mothers step by step, which can provide a more comprehensive welfare program, such as accelerate the implementation of paternity leave and parental leave, thus can stimulate enthusiasm for father to participate in childcare and offer practical help. Additionally, our study shows that the scores of support from people the mother can rely on to help meet emotional needs were low, a possible explanation might family members were given insufficient attention to lactation mothers in recent years, according
**Table 2 Variable assignments.**

| Variables | Risk factors | Assignment |
|---|---|---|
| X1 | Age | 20∼29 = 1, 30∼35 = 2, >35 = 3 |
| X2 | Ethnic groups | The Han nationality = 1, the other nationality = 2 |
| X3 | Highest level of schooling | Junior high = 1, high school = 2, junior college education = 3, university = 4, postgraduate or higher = 5 |
| X4 | employment status | Be unemployed = 1, be employed = 2 |
| X5 | Financial status | <3000 = 1, 3000∼4999 = 2, 5000∼7999 = 3, ≥8000 = 4 |
| X6 | The most recent delivery mode | Natural labor = 1, caesarean section = 2, vaginal delivery = 3 |
| X7 | Number of babies | 1 = 1, 2 = 2, ≥3 = 3 |
| X8 | Gender of the youngest baby | Male = 1, female = 2 |
| X9 | The age of the youngest baby | 0∼5 months = 1, 6∼12 months = 2, 13∼24 months = 3 |
| X10 | The feeding mode of the youngest baby | Exclusive breastfeeding = 1, mixed feeding = 2 |
| X11 | The way of confinement | Family members = 1, confinement nurse = 2, confinement center = 3, nanny = 4 |
| X12 | Whether or not breastfeeding at night | Yes = 1, no = 2 |
| X13 | Main caregiver of the youngest baby during the day | Grandmother/grandfather = 1, own = 2, father = 3, nanny/confinement nurse = 4 |
| X14 | Maternity leave time | No = 1, Below statutory maternity leave = 2, Statutory maternity leave (158∼173 days) = 3, Above statutory maternity leave = 4, jump = 5 |
| X15 | Whether or not return to work | Yes = 1, no = 2, jump = 3 |
| X16 | Whether or not have lactation room in the workplace | Yes = 1, no = 2, jump = 3 |
| X17 | Mother attended school/lecture for pregnant women | Never = 1, 1∼3 = 2, 4∼6 = 3, >7 = 4 |
| X18 | Partner attended school/lecture for pregnant women | Never = 1, 1∼3 = 2, 4∼6 = 3, >7 = 4 |
| X19 | Community promotion of breastfeeding methods | Bulletin board = 1, phoned or texted or answered questions = 2, provided house calls = 3, no = 4 |

**Table 3 Risk factors selected by LASSO regression.**

| Variables | Risk factors | Coefficient |
|---|---|---|
| X12 | Whether or not breastfeeding at night | −2.81344729 |
| X4 | Employment status | −0.31132352 |
| X10 | The feeding mode of the youngest baby | −0.17325848 |
| X18 | Partner attended school/lecture for pregnant women | 0.09578978 |

to China's seventh national population census, the total fertility rate has fallen below 1.3, and the overall population declined by 850,000 (*China Government Network, 2021*). Along with the decreased population of women of childbearing, the delayed age of marriage and childbearing, the change of ideas about having child, family members might more focus on the process of pregnancy, in contrast, ignore the process of breastfeeding.

Our investigation shows that whether or not breastfeeding at night is the most important factor in breastfeeding social support, the level of social support for mothers who breastfeed at night is higher than that of mothers who do not breastfeed at night. This corresponds with the finding of Aerts's study (*Aerts, Janaqi & Cock, 2023*), in which it demonstrated that

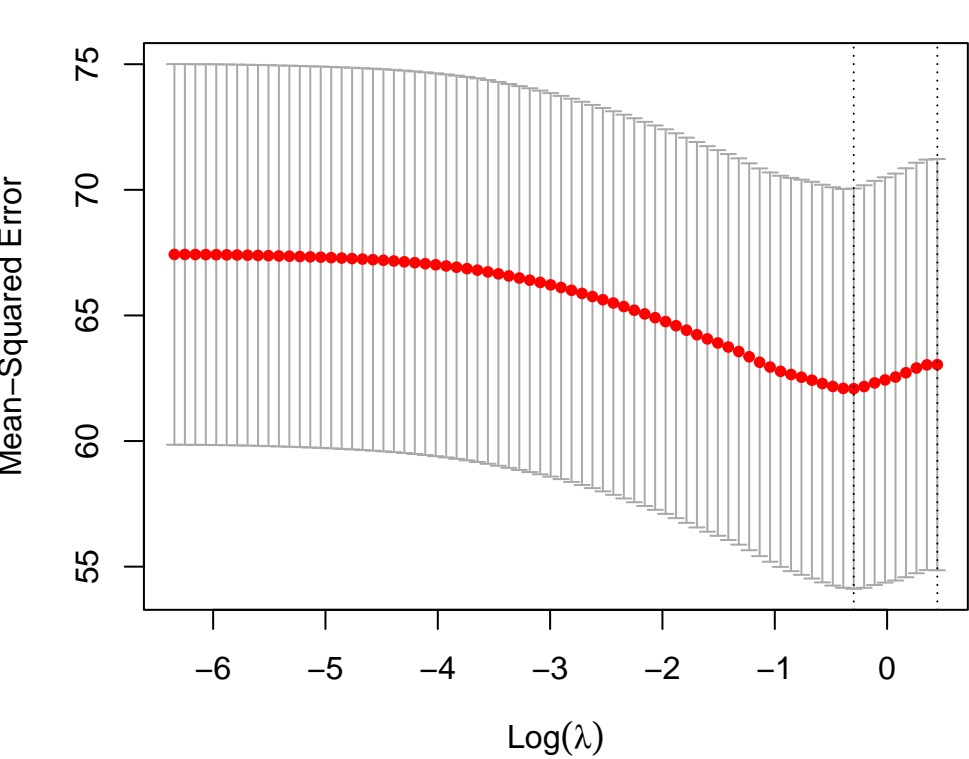

**Figure 2** The LASSO regression results.

fathers taking part in breastfeeding care during the night would positively impact mothers' N3 (slow-wave) sleep duration and milk production. *Tikotzky, Sadeh & Glickman-Gavrieli (2011)* found that mothers who breastfeed at night perceived less support from their partner, which indirectly mirrors the significance of social support, especially partner support, during mothers' breastfeeding at night. Practically, breastfeeding at night requires additional effort, as mothers have to awaken from restful sleep repeatedly (*Mirzakhmetova et al., 2024*). During this time, the family members acted as timely facilitators, providing crucial support and assistance. With the support from family members, mothers may feel that they are not facing challenges alone, contributing to their active breastfeeding practice (*Mörelius et al., 2021*). However, the mothers who did not breastfeed at night may not adequately experience the support and empathy from their family members to some extent, which influences their perceived social support level, and may explain the score differences between the mothers whether or not breastfeeding at night. This highlights that health care providers should be concerned about the mothers who do not breastfeed at night, and encourage their partner to share nursing work for babies during the night. At the national level, it is also recommended that further policy support be provided to lactating maternal partners, so that they can have enough energy to participate in child-rearing activities at night.
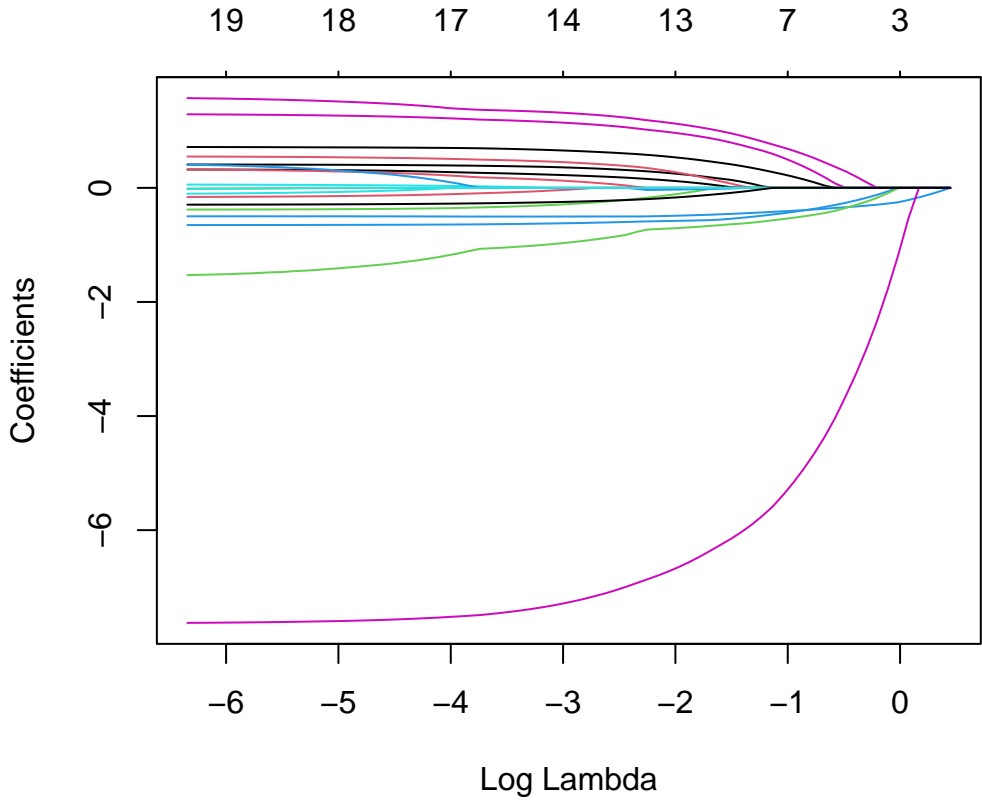

**Figure 3** LASSO coefficient profiles of the four variables.

Our study also suggests that employment status is the second factor in breastfeeding social support; the level of social support for mothers who are unemployed is higher than that of mothers who are employed, which is consistent with a previous study conducted in Australia (*Ayton et al., 2025*). In China, if mothers are unemployed, they usually stay at home as housewives, so they tend to have more time and energy to communicate with their family members, relatives and friends, and obtain their support and help in terms of breastfeeding (*Gebrekidan et al., 2021*).

Conversely, employed mothers need to deal with the work-family conflict, which may burn their capacity to perceive the support around them. For the influence of traditional Chinese culture, mothers, as primary caregivers, are asked to dedicate more effort to family. They have to bear the feeding pressure, such as expressing, storing and transporting breast milk. At the same time, they must be a strong woman in the workplace, running around making achievements to hold down their job (*Wolde, Ali & Mengistu, 2021*). Furthermore, the lack of workplace support may be another cause of the lower scale scores for employed mothers, which includes the absence of breastfeeding culture and facilities in the workplace, such as the friendly policy for breastfeeding breaks and flexible working hours, as well as the lactation rooms and fridges for mothers to pump and save breast milk (*Rowbotham et al., 2022*; *Burns et al., 2023*). However, the sources of social support of the Chinese version

of BSSS include family members, health care providers, close friends, peers and others, not taking into account the targeted support for the workplace, but some items may concern it. Nonetheless, This conclusion is different from the study by *Dongdong (2022)*, it is concluded that there is no discernible difference in the level of breastfeeding social support between unemployed and employed mothers. It may be that the ways of classifying occupation in the two studies are different.

The feeding mode of the youngest baby is the third factor influencing breastfeeding social support in our study, mothers who exclusively breastfed their infants exhibited higher levels of social support for breastfeeding than mothers who engaged in mixed feeding. This is in agreement with the finding of *Zeng et al.*'s (*2024*) study. This may because the mothers who mix breastfeeding may suffer greater challenges than those who breastfeed exclusively. Mothers who mix breastfeeding could feel more embarrassment, tiredness and problems (*Soltani et al., 2024*). They might even experience guilty and overload due to they could not exclusively breastfeed their infants (*Alves et al., 2023*). Thus, they require more social support while breastfeeding, and it appears to be less social support than exclusive breastfeeding mothers. Yet a study by *Dongdong (2022)* found no significant difference in the level of maternal breastfeeding social support across different infant feeding styles. This may be a consequence of the differing survey instruments and survey populations, so further research is required to explore the relationship between different infants' feeding styles and the level of breastfeeding support in order to inform healthcare professionals' targeted intervention strategies.

In this study, partner attended school /lecture for pregnant women was also identified as a factor that influences breastfeeding social support, mothers whose partners had got training in maternity school or attended prenatal educational sessions demonstrated higher levels of breastfeeding social support than those whose partners had no experience of such training. In addition, our investigation revealed that the majority of partners in this study (88%) had not attended a maternity school or prenatal education sessions, which is consistent with a study by *Xuejun (2021)*, it is discovered that as many as 85.2% of Pregnant women's partners had not attended maternity school before. However, Pregnant women's partners participation in maternity school had a positive effect on the level of breastfeeding support provided by partners. Thus, it is important to encourage pregnant women's partners to attend prenatal educational sessions, lecture or school for improving breastfeeding.

To our knowledge, this study is the first investigation of breastfeeding social support among the *puerpera* using the Chinese version of the Breastfeeding Social Support Scale. Besides, with the data analysis of the lasso regression, the factors associated with breastfeeding social support can be effectively evaluated and sorted by importance. This can assist health care providers develop appropriate interventions from the aspect of postnatal breastfeeding experience, mother socioeconomic characteristics and breastfeeding promotion policies, thereby increasing breastfeeding rates and duration of breastfeeding.

There are some limitations to this study. Firstly, our study was conducted in four community health service centers in Chengde, Heibei province, China, so generalizability across ethnic groups in other areas or countries should proceed with caution. Secondly, it is a

cross-sectional study, the data can only show the correlations in nature, not imply causation. Additionally, we don't have a standard for evaluating the level of breastfeeding social support, which hindered us further exploring classification standards for the scale's level. Furthermore, among the factors considered, the analysis was limited in basic information and the impact of some related breastfeeding polices, factors such as the fertility allowance, the applications for government funding for milk powder and the breast milk repository were not included in our analysis, because they were not available in Chengde City. This suggests further research is required to assess various elements about breastfeeding social support.

## CONCLUSIONS

In this study, the breastfeeding social support in Chengde city is positioned at an upper-middle level, the independent influencing factors for the breastfeeding social support included whether or not breastfeeding at night, employment status, the feeding mode of the youngest baby, partner attended school for pregnant women/lecture, which can assistant health care providers develop targeted interventions to improve breastfeeding rates according to influencing factors.

## ACKNOWLEDGEMENTS

The authors are grateful the support of all the community health service centers in Chengde City, we also thank all the mothers who participated in this study.

### Funding

The study was funded by the Student Innovation and Entrepreneurship Training Program of Chengde Medical College (2023068) and the Social Science Development Project of Chengde City (20223112). The funders had no role in study design, data collection and analysis, decision to publish, or preparation of the manuscript.

### Grant Disclosures

The following grant information was disclosed by the authors:
The Student Innovation and Entrepreneurship Training Program of Chengde Medical College: 2023068.
The Social Science Development Project of Chengde City: 20223112.

### Competing Interests

The authors declare there are no competing interests.

### Author Contributions

- Tian Sun conceived and designed the experiments, performed the experiments, analyzed the data, prepared figures and/or tables, authored or reviewed drafts of the article, and approved the final draft.

- Chanzhi Duan analyzed the data, prepared figures and/or tables, and approved the final draft.
- Yan Wang performed the experiments, authored or reviewed drafts of the article, and approved the final draft.
- Qing Li conceived and designed the experiments, authored or reviewed drafts of the article, and approved the final draft.

## Human Ethics

The following information was supplied relating to ethical approvals (*i.e.*, approving body and any reference numbers):

Our study has been approved by the Medical Ethics Committee of Chengde Medical College (number: CYFYLL2022331).

## Data Availability

The data is available in the Supplemental File.

## Supplemental Information

Supplemental information for this article can be found online at http://dx.doi.org/10.7717/peerj.18779#supplemental-information.

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
