# Peer review of "The status quo and influencing factors of breastfeeding social support in China based on LASSO regression model"

_PeerJ, doi:10.7717/peerj.18779_

## Round 0.1 · original submission · Minor Revisions

Please follow carefully all comments and recommendations.

·

Basic reporting

No comment

Experimental design

No comment

Validity of the findings

No comment

Additional comments

In general, the manuscript is good. However, there are several explanations that must be added in the following discussion section.

Please give logic explanation about the relationship between breastfeeding at night and breastfeeding social support. Why the level of social support for mothers who breastfeed at night is higher than that of mothers who do not breastfeed at night? (line 165-167)

Please give logic explanation about the relationship between employment status and breastfeeding social support. Why the level of social support for mothers who unemployed is higher than that of mothers who employed? What is the scope of your social support? Your explanation discusses about workplace support. (line 177-186)

Please give logic explanation about the relationship between the feeding mode of the youngest baby and breastfeeding social support. Why mothers who exclusively breastfed their infants exhibited higher levels of social support for breastfeeding than mothers who engaged in mixed feeding? (line 189-190)

Reviewer 2 ·

Basic reporting

good paper but needs revisions

Experimental design

none

Validity of the findings

it is OK

Additional comments

-Make sure that each paragraph at least contains three sentences.
- Each reference must be completed with DOI and can be traced online.
- some references are old, please update in recent 5 years
Discuss the results/findings with more references to make them meaningful. Discuss and compare your findings with previous studies and or grand theories. The current discussion section is insufficient with fewer references; please improve and enlarge. Support each paragraph with at least one reference.
In the Method section: explain how you determined the sample size. Support with the reference whether the sample size is adequate.
Explain the validity and reliability of your questionnaire/instrument
How did you control the confounding variables?

---

## Round 0.2 · accepted · Accept

Thank you for your contribution!

·

Basic reporting

Everything is good

Experimental design

Everything is good

Validity of the findings

Everything is good

Additional comments

All the input that has been given before has been corrected and this article can be published